# Cerebral Aneurysms Caused by Atrial Myxoma—A Systematic Review of the Literature

**DOI:** 10.3390/jpm13010008

**Published:** 2022-12-21

**Authors:** Justyna Chojdak-Łukasiewicz, Sławomir Budrewicz, Marta Waliszewska-Prosół

**Affiliations:** Department of Neurology, Wroclaw Medical University, 50-556 Wroclaw, Poland

**Keywords:** atrial myxoma, cerebral aneurysm, metastatic aneurysm, headache, cardiac tumors

## Abstract

**Background**: The association between cerebral aneurysms and left atrial myxoma is known but rare. We described its pathogenesis, clinical presentation, diagnostic findings and treatment using a systemic review of the literature. **Methods**: MEDLINE via PubMed was searched for articles published until August 2022 using the keywords “atrial myxoma”, “cardiac myxoma” and “cerebral aneurysm”. **Results**: In this review, 55 patients with multiple myxomas aneurysms were analyzed, and 65% were women. The average age when aneurysms were diagnosed was 42.5 ± 15.81; most patients were less than 60 years old (86%). Aneurysms could be found before the diagnosis, at the same time as cardiac myxoma, or even 25 years after resection of the atrial mass. In our review, the mean time to diagnoses was 4.5 years. Our review estimates that the most common symptoms were vascular incidents (25%) and seizures (14.3%). In 15 cases, variable headaches were reported. Regarding management strategies, 57% cases were managed conservatively as the primary choice. **Conclusions**: Although cerebral aneurysms caused by atrial myxoma are rare, the long-term consequences can be serious and patients should be monitored.

## 1. Introduction

Cardiac myxomas (CM) are the most common benign “cardiac” tumors, accounting for up to 30–50% of all primary heart tumors [1]. The incidence is approximately 0.5–1 cases per 1,000,000 population per year [2]. About 75% concern the left atrium of the heart [3], and 18% originate in the right atrium; biatrial myxomas are rare and account for less than 2.5% of all cardiac myxomas. Myxomas are particularly frequent from the third to the sixth decades of life; the ratio women: men varies from 2:1 to 3:1. CM are diagnosed based on clinical examination and tests such as electrocardiography (ECG), transthoracic echocardiogram (TTE), transesophageal echocardiogram (TEE), chest computed tomography (CT) or magnetic resonance imaging (MRI) and cardiac MRI [4].

Most cardiac myxomas present with constitutional, embolic and obstructive manifestations. Younger and male patients have more neurologic symptoms, and female patients have more systemic symptoms. These can cause many neurological complications, including systemic embolism, cerebral infarction, cerebral cavernous malformations and intracranial aneurysms [5]. Myxoma-related aneurysms are always multiple and in most cases have a fusiform-shape.

Left atrial myxomas are considered curable by complete resection and give excellent results in long-term follow-up. Surgical excision remains the treatment of choice for cardiac myxoma. Early diagnosis and intervention is desirable because of the persistent risk of brain metastases and aneurysms. However, incomplete resection, multifocal tumors and embolism caused by tumors are important factors in its recurrence and complications [3,6]. Currently, our understanding of cerebral aneurysms caused by atrial myxoma is based mainly on case reports.

This systemic review of the literature aimed to provide an exhaustive summary of available case reports evaluating medical history, clinical, diagnostic and therapeutic methods in patients with cerebral aneurysms caused by atrial myxoma.

## 2. Methods

JCŁ and MWP performed an independent online search in accordance with PRISMA guidelines [7] using the following combination of keywords: “atrial” and “cardiac” and “myxoma” and “cerebral” and “aneurysm” or “myxomatosus” and “cerebral” and “aneurysm”.

We considered publication records from MEDLINE and ERIC databases until August 2022. In addition, the reference lists from eligible publications were searched. All discrepancies were resolved by discussing the results of the preliminary search with a third reviewer (SB) (Figure 1).

A total of 257 records were identified and screened separately by the authors. Then, these record lists were double read by both analysts and 92 abstracts were found to be relevant to the subject. Each researcher worked independently and prepared their own list of relevant full-text manuscripts. Both lists were compared and 54 publications were found to be the most relevant to the study and included in this review. The exclusion criteria were non-English-language articles, conference papers and abstract only.

## 3. Results

We found 54 case report articles describing 55 patients. All cases are illustrated in Table 1. We did not find any article about type case series or an original work on a larger group of patients. The group consisted of 35 women (64%) and 20 men (36%). The average age when aneurysms were diagnosed was 42.5 ± 15.81 years (the age varied from 11 to 69 years) and 46 of the patients were less than 60 years old (84%).

Aneurysms could be found before the diagnosis, at the same time as cardiac myxoma, or even 25 years after resection of atrial mass. In our review, the mean time to diagnoses was 4.5 years. In 1 patient, the myxoma was localized in both atrial, while in the remaining 54 patients—left atrium.

Our review estimates that the most common symptoms were vascular incidents (TIA, stroke), seizures, vertigo or dizziness and loss of consciousness. In 15 cases, variable headaches were reported—most often they had the migraine phenotype with visual disturbances. Three patient presented clinical symptoms typical of subarachnoid hemorrhage and two had no symptoms (Table 2).

Based on our analyses, trial myxoma-associated aneurysms are most often localized to the entire area of vascularization, followed by middle cerebral arteries, posterior cerebral arteries, anterior cerebral arteries and finally the basilar artery (Table 3).

All patients underwent successful surgical resection of the cardiac myxoma. Regarding management strategies, 33 patients (60%) were managed conservatively as the primary choice. In three cases (5.5%) there was chemotherapy treatment; in one case, radiotherapy. One patient was treated with stereotactic radiosurgery.

## 4. Discussion

Cardiac myxomas are the most common benign cardiac tumor in adults [58]. Myxoma cells most likely arise from resident pluripotent or multipotent mesenchymal stem cells, the embryonic remnants of which differentiate into endothelial cells, smooth muscle cells and other mesenchymal cells and this explains the most common occurrence of myxomas in the atrial septum [58,59]. Myxomas of the heart are most common in adults between the third and sixth decades of life. They can occur sporadically (more often in women) or be familial [1,2,58]. Familial occurrence has been shown to be associated with an autosomal dominant mutation of the PRKAR1A gene located on chromosome 17q2 [59,60]. Familial myxomas are usually multiple, recurrent and located outside the left atrium [61].

The genetic basis of intracranial aneurysms is very complex. In recent years, there has been a growing interest in the extracellular matrix surrounding cerebral vessels, as well as the role of matrix metalloproteinases [62,63]. Studies on the genetics of aneurysms have been aimed at elucidating causative genes or discovering new loci associated with aneurysm risk. Genome-wide association studies have used single nucleotide polymorphism data to discover several susceptibility loci, including the SOX17 and CDKN2A genes. The proteins encoded by these genes regulate endothelial function and blood vessel formation and so genetic variation that affects the extracellular matrix may have the greatest impact on the risk of aneurysms [62,64].

The clinical picture of CM includes symptoms due to embolism, intracardiac obstruction, size, location and mobility of the tumor [1,59]. Patients with small tumors may remain asymptomatic for years, or nonspecific symptoms may mimic systemic or cardiovascular disease [58].

Embolism associated with detachment of tumor fragments or thrombi occurs in 10–50% of patients with cardiac myxomas [59,65]. There has been no correlation between the risk of embolism and tumor size and some authors have suggested an association of such complications with chest trauma [66]. Most commonly, embolisms involve the cerebral arteries, where cavernous malformations and aneurysms can develop. Neurological complications are a very broad group of symptoms that include fainting and loss of consciousness, headache and dizziness, seizures, transient cerebral ischemia, stroke or rupture of aneurysms or vascular malformations [5,8,51]. Women in their fifth decade of life are most at risk for embolic stroke and acute embolic stroke may be the first manifestation of atrial myxoma in a young patient [58,59]. Sudden loss of consciousness after strenuous exercise is particularly important in the patient’s history [58]. Embolism of the coronary artery is rare, and it is even believed that the coronary arteries are relatively resistant to embolism due to anatomical conditions [1,65,67]. Braun et al. [61] showed from their analysis that only 40 cases of myocardial infarction due to myxoma have been documented in the literature.

The etiology of myxomatous cerebral aneurysms is still unknown [40,59]. A few hypotheses have been put forward in terms of the pathogenesis of aneurysms. Based on the literature, two main theories can be identified. First, a neoplastic process theory proposes that myxoma cells adhere to and penetrate the endothelium, then grow in the subintimal layer and destroy the arterial wall. However, it should be remembered that the metastatic hypothesis does not imply a typical tumor metastasis process. By definition, cardiac myxomas are not malignant tumors and therefore do not have potential to “metastasize” in the strict sense of the word. The second theory is the “vascular damage theory” proposed by Sloane et al. in 1966, where the temporary occlusion of cerebral vessels by myxoma cells causes damage to the endothelium, which is followed by an alteration of hemodynamics and promotion of aneurysm formation [68,69,70].

Myxoma cells produce and release proinflammatory cytokine interleukin-6 (IL-6), which is an important factor of aneurysm initiation [18]. Recent studies suggest that autocrine production of IL-6 by myxoma plays a main role in the embolization of the myxomatous cell. Elevated IL-6 levels have been detected in patients with myxomatous aneurysms, before and even after myxoma resection. It has been known that atrial myxoma cells are capable of producing IL-6. Recent studies have shown that there is a connection between overproduction of Il-6 and cerebral aneurysm development. A persistent elevated IL-6 level induces overexpression of multiple proteolytic enzymes (such as metalloproteinase), which can weaken cerebral vessel walls and lead to aneurysm formation [55,71]. Based on this theory, cardiac myxoma resection is usually accompanied by a reduction in serum IL-6 levels, but a few studies have shown new aneurysm formation after the myxoma resection still showed persistently elevated IL-6 levels. Formation of a cerebral aneurysm is also associated with overproduction of Il-6 by an emboli tumor, that induces degradation of the extracellular matrix in the intracranial vessels and is connected with an increased level of IL-6 in cerebrospinal fluid. So, IL-6 has two ways of impacting the formation of the aneurysm’s direction, first by promoting tumor invasion into the intracranial artery or secondly by increasing the chance of a distant embolization of the cardiac myxoma [55,57,71].

The natural history of this kind of aneurysm is also not clear; some cases have shown stability, others have shown improvement (self-occlusion) and others have shown an increased number and enlargement of aneurysms [5,51,57]. In most cases, the first neurological manifestation of atrial myxomas is complications due to cerebral embolism and subsequent cerebral infarction [2,59,65]. Vascular incidents (transient cerebral ischemia, stroke), which were observed in 36.3% of the patients of this review, should be precisely associated with embolism. Aneurysm formation and subsequent subarachnoid or intracerebral hemorrhage are rare but are the most well-known complications of atrial myxoma in adults [5]. Even the presence of multiple but unruptured cerebral vascular aneurysms usually does not produce clinical symptoms. However, up to half of patients with cerebral aneurysms may experience so-called “predictive headaches,” the exact cause of which is not known, but is thought to be related to microbleeding from aneurysms or other vascular malformations [69,72,73]. The other symptoms that were observed in the patients analyzed in this review included seizures, headaches or dizziness that could be related to microbleeding from aneurysms or could be a symptom related to compression of malformations on central nervous system structures. Given some nonspecific but nevertheless quite suggestive clinical signs, it is necessary to screen for cerebral complications in patients with atrial myxoma.

Currently, there are no guidelines for the treatment of aneurysms caused by cardiac myxomas, but a conservative approach and radiological follow-up is recommended. The majority of reported cases have demonstrated stability and some have even been documented as exhibiting spontaneous regression [12]. Routine radiological follow-up by MRI examination is needed to monitor the eventual progression of the aneurysms [72,73]. A lot of therapeutic methods are available, ranging from endovascular methods, surgery, chemotherapy, radiation or a combination of these. Only enlarged or ruptured aneurysms may require invasive management and must be evaluated for endovascular or neurosurgical intervention [4].

The atrial myxoma should be excised as soon as possible after the diagnosis to prevent further complications such as systemic embolization, constitutional symptoms (fever, fatigue, weight loss) or obstruction of the mitral valve [44,58,59]. Surgical resection of the cardiac myxoma also eliminates the early neurologic symptoms, most frequently ischemic cerebral infarcts. Although the cardiac resection of the atrial tumor minimizes the risk of embolization, it does not decrease the risk of the formation of a delayed cerebral aneurysm. This results from the theory of “metastasis and infiltrate”. Intracranial aneurysms may continue to grow despite the surgical removal of the atrial myxoma [53,67].

The current literature describes several different surgical options. Cases of ruptured aneurysms are generally considered as urgent surgical procedures. Clipping or coiling are not applicable for myxomatous aneurysms because they are multiple, located at distal vessels, fusiform and without a neck. The literature provides a few reports about clipping of large aneurysms [54]. Aneurysms might keep growing after endovascular coil embolization [31].

Open surgical treatment is recommended for a lesion-caused mass effect or in cases of single saccular aneurysms. A bypass is recommended for lesions with good collateral compensation and it is a reasonable option when sacrifice of the feeding artery may be required. Compared with other options, this procedure is technically challenging and is limited because it is difficult to apply in a variety of locations where aneurysms may occur [33].

Chemotherapy as a treatment was introduced by Roeltgen et al. in 1981 [74]. They tried doxorubicin in conjunction with surgery for recurrent atrial myxoma. In some cases, etoposide and carboplatin were also used [13,50]. Chemotherapy may protect patients against aneurysm growth [13]. Low-dose radiation in combination with chemotherapy has been reported as an effective method for degradation of metastasis [5,12,13]. A new option is frameless stereotactic radiosurgery (SRT), which is less invasive than endovascular or open surgery, avoids the systemic effects of chemotherapy, and limits toxicity to surrounding brain parenchyma compared to whole brain irradiation [5].

## 5. Conclusions

Cerebral aneurysms are rare complications of cardiac myxoma, which can appear many years after cardiologic treatment. They are twice as common in middle-aged women. The entire area of vascularization is most often located in the area of the middle cerebral artery. Vascular incidents, unspecific headaches and seizures are their most common clinical manifestations; their rupture and subarachnoid hemorrhages are relatively rare. We do not have any treatment guidelines as yet, however, in the case of myxoma aneurysms a long-term observation is recommended.

Therefore, long-term follow-up of patients with cardiac myxomas for possible co-occurrence of cerebral aneurysms and their complications is very important. In addition, patients with multiple cerebral aneurysms, especially those with a cardiac burden, should be alert to the possibility of cardiac myxoma.

## Figures and Tables

**Figure 1 jpm-13-00008-f001:**
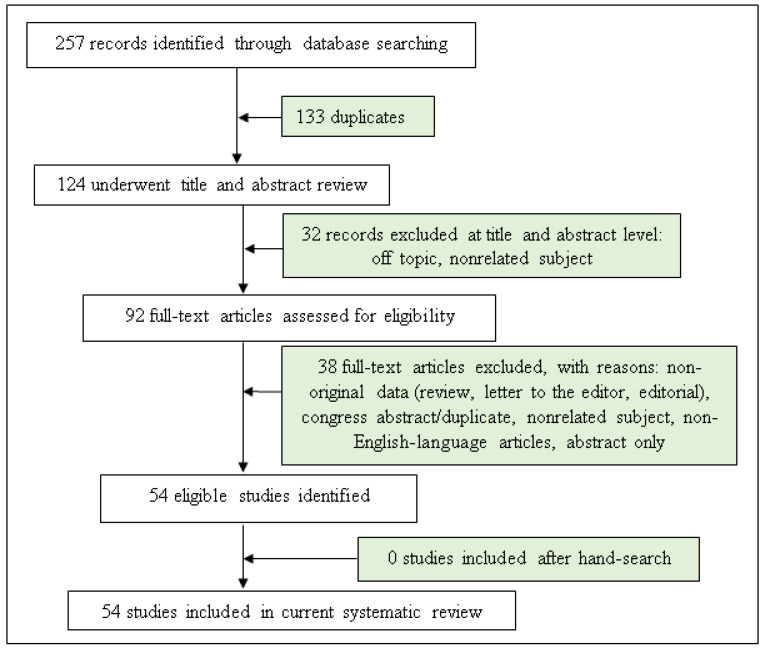
Flow chart of study selection.

**Table 1 jpm-13-00008-t001:** Overview of cases of cerebral aneurysms in atrial myxoma from the literature.

	Author	Case	Clinical Presentation	Atrial Myxoma History	Radiological Findings	Cardiological Treatment	Aneurysm Procedure
1.	Alrohimi et al. [8]	37-year-old woman	thunderclap headache with right-sided ptosis	left atrial myxoma diagnosed at the same time	CTA—right posterior communicating artery aneurysm	open heart surgery	right posterior communicating aneurysm clipping
2.	Asranna et al. [9]	57-year-old woman	secondary generalized seizures	left atrial myxoma resection 1 year earlier	DSA—multiple fusiform aneurysms involving the left middle cerebral artery (MCA) M3 segment and angular branch	open heart surgery	conservative
3.	Ashalatha et al. [10]	54-year-old man	left focal motor seizures with secondary generalization	left atrial myxoma 6 months earlier	DSA—multiple, small, distal, fusiform aneurysms along both middle and anterior cerebral arteries	open heart surgery	conservative
4.	Baikoussis et al. [11]	72-year-old woman	vertigo and collapse with loss of consciousness	left atrial myxoma diagnosed at the same time	MR—multiple cerebral mycotic aneurysms of various dimensions and a large cyst, as a result of a previous hemorrhage	open heart surgery	embolization of the large central aneurysms
5.	Bernet et al. [12]	31-year-old woman	general clonic–tonic seizure	left atrial myxoma resection 2 months earlier	CT—multiple frontal and occipital bilateral cerebral aneurysm	open heart surgery	radiation plus chemotherapy
6.	Branscheidt et al. [13]	41-year-old woman	‘‘burning’’ headaches and increasing fatigue	left atrial myxoma diagnosed at the same time	DSA—multiple fusiform aneurysms	open heart surgery	chemotherapy
7.	Chen et al. [14]	19-year-old woman	seizures without loss of consciousness	left atrial myxoma resection 2 years earlier	DSA—many saccular dilations on the distal end of the MCA and PCA of both sides	open heart surgery	conservative
8.	Chow et al. [15]	58-year-old woman	loss of consciousness 2 years earlier SAH (external ventricular drainage was executed)	left atrial myxoma without resection	DSA—lobulated aneurysm at the middle cerebral artery with clipping executed.	open heart surgery	conservative
9	Desousa et al. [16]	44-year-old woman	left-sided headache, vomiting	left atrial myxoma resection 8 years earlier	carotid angiogram demonstrated progressive narrowing of the left internal carotid artery	open heart surgery	conservative
10.	Eddleman et al. [17]	18-year-old man	episode of scintillations in the right visual field lasting 2 h associated with a headache	left atrial myxoma resection 4 months earlier	DSA—multiple fusiform aneurysms the distal anterior, middle, and posterior circulations	open heart surgery	resection some aneurysm
11.	Ezerioha et al. [18]	73-year-old woman	SAH	left atrial myxoma recognized at the same time	CTA 9-mm lobulated aneurysm at the right middle cerebral artery (MCA) trifurcation and a small 3-mm aneurysm at the left MCA bifurcation	open heart surgery	right frontal pterional craniotomy, evaluation of intracerebral hematoma and clipping of the right MCA aneurysm
12.	Flores et al. [19]	19-year-old woman	right-sided hemiparesis lasting for one hour	left atrial myxoma recognized at the same time	DSA—multiple fusiform cerebral aneurysms affecting several distal branches of both middle cerebral arteries	open heart surgery	conservative
13.	Flores et al. [19]	61-year-old man	acute onset of rotational vertigo and left visual field deficit, stroke 20 years ago	left atrial myxoma recognized at the same time	DSA—multiple fusiform cerebral aneurysms in the left posteroinferior cerebellar artery, two aneurysms in the M2 segment of the right middle cerebral artery	open heart surgery	conservative
14.	Furuya et al. [20]	36-year-old man	sudden attack of generalized convulsive seizures	left atrial myxoma resection 1 year earlier	DSA—multiple fusiform aneurysms at right operculofrontal, central, and angular arteries	open heart surgery	resection aneurysms of the angular artery
15.	George et al. [21]	45-year-old woman	transient ischemic attack	left atrial myxoma recognized at the same time	DSA—multiple fusiform aneurysm at the right middle cerebral artery aneurysm	open heart surgery	conservative
16.	Gupta et al. [22]	11-year-old boy	syncope	left atrial myxoma resection 1 year earlier	CT angiography tortuous, dilated and fusiform left MCA and multiple aneurysms in bilateral MCA and both vertebral arteries	open heart surgery	conservative
17.	Herbst et al. [23]	31-year-old man	dizziness, nausea, blurred vision of his left eye, and gait disturbance	left atrial myxoma was discovered at the same moment	DSA—multiple intracranial microaneurysms in peripheral branches of middle, anterior, and posterior cerebral arteries; a few aneurysms were seen in branches of the vertebrobasilar arteries	open heart surgery	conservative
18.	Hau et al. [5]	57-year-old man	confusion and memory loss	left atrial myxoma resection 2 years earlier	CTA—multiple fusiform intracranial aneurysms at left anterior cerebral artery (ACA) A2 segment bifurcation, right middle cerebral artery (MCA) distal M2 segment, cortical branches at frontal and para-central regions, left posterior cerebral artery (PCA) P3 segment, and right occipital cortical branches, with progressive enlargement half-yearly	open heart surgery	stereotactic radiosurgery
19.	Iskandar et al. [24]	69-year-old woman	left arm numbness, weakness, and dysarthria	left atrial recurrent myxoma resection 20 and 15 years earlier	CTA—myxomatous fusiform aneurysms in the right middle and anterior cerebral arteries	open heart surgery	conservative
20.	Ivanovic et al. [25]	44-year-old woman	ten months earlier SAH with operation of left PICA aneurysm	left atrial myxoma ten months earlier	DSA—saccular aneurysm arising from the origin of left posterior inferior cerebelli artery	open heart surgery	conservative
21.	Jean et al. [26]	32-year-old woman	transient ischemic attack	left atrial myxoma resection 5 years earlier	DSA—multiple peripheral, fusiform, intracranial aneurysms	open heart surgery	left frontal craniotomy for resection of one of the aneurysms located at the frontal pole
22.	Josephson et al. [27]	33-year-old woman	8 years earlier multiple embolic strokes	left atrial myxoma resection 8 years earlier	MRA—multiple fusiform aneurysms	open heart surgery	conservative
23.	Kim et al. [28]	58-year-old woman	right flank pain for several days 20 years earlier three episodes of stroke with dysarthria and right-sides hemiplegia	left atrial myxoma discovered at the same moment	MRA—multiple fusiform aneurysms of the left distal internal carotid artery, peripheral branch of the right middle cerebral artery, left posterior cerebral artery, and the distal basilar artery	open heart surgery	conservative
24.	Koo et al. [29]	64-year-old woman	dysarthria, generalized weakness, and gait disturbance	left atrial myxoma discovered at the same moment	DSA—multiple fusiform-cerebral aneurysms at distal branches of anterior cerebral arteries (ACA) and middle cerebral arteries (MCA)	open heart surgery	conservative
25.	Krishnan et al. [30]	31-old-year man	two episodes of generalized tonic clonic seizures	left atrial myxoma resection 12 years earlier	CTA—fusiform dilation of bilateral distal anterior cerebral arteries, multiple dilations of distal middle cerebral artery branches on both sides and also aneurysmal dilatation of the distal right posterior cerebral artery	open heart surgery	conservative
26.	Lazarow et al. [31]	52-year-old man	acute right lower extremity weakness and seizures	left atrial myxoma resection 3 years earlier	DSA—diffuse cerebral arterial aneurysms	open heart surgery	left MCA branch was embolized with aneurysm coils
27.	Li et al. [32]	27-year-old woman	sudden onset of vertigo, dysarthria and right-sided weakness	left atrial myxoma recognized at the same time	DSA—multiple typical distal fusiform and saccular aneurysms or aneurysmal dilations in the bilateral internal carotid artery territories	open heart surgery	conservative
28.	Namura et al. [33]	45-year-old man	right hemiparesis 10 years earlier	left atrial myxoma resection after 10 years	DSA—multiple cerebral aneurysms	open heart surgery	conservative
29.	Oguz et al. [34]	40-year-old man	numbness in right arm and blurred vision	left atrial myxoma resection 5 years earlier	DSA—fusiform dilatations in the prefrontal branch of the right MCA, the angular and frontal branches of the left MCA, and the calcarine branch of the left vertebral artery	open heart surgery	conservative
30.	Oomen et al. [35]	40-year-old woman	sensory loss in tongue and face, and word finding difficulty	left atrial myxoma resection 1 year earlier	DSA—micro-aneurysms in the right middle cerebral artery	open heart surgery	conservative
31.	Quan et al. [36]	49-year-old man	acute headache and dizziness	left atrial myxoma recognized at the same time	MRA—multiple small aneurysms	open heart surgery	conservative
32.	Penn et al. [37]	12-year-old boy	a sudden headache, diplopia, gait instability, and speech difficulty	left atrial myxoma recognized at the same time	DSA—numerous a flame-shaped or fusiform dilation on the right internal carotid artery (ICA), a sausage-like fusiform dilation of the right posterior cerebral artery (PCA)	open heart surgery	endovascular treatment
33.	Radoi et al. [38]	45-year-old man	headache, nausea, gait disturbances and weakness of the left extremities	left atrial myxoma resection 16 months earlier	DSA—multiple unruptured intracranial microaneurysms, which were mainly located in the peripheral branches of the left anterior and middle cerebral arteries	open heart surgery	resection the right parietal lesion
34.	Ryou et al. [39]	27-year-old woman	sudden onset dizziness, headache, blurred vision, and tingling sensations in tongue, arm, and the left side of her face	atrial myxoma on both sides, resection 10 years earlier	DSA—revealed multiple fusiform aneurysms in the basilar artery, proximal PICA, left P2 and right P4 segments, temporal branch of the left MCA, and distal branches of the right MCA and ACA	open heart surgery	conservative
35.	Sabolek et al. [40]	43-year-old woman	sudden severe headache, nausea, consciousness disturbances	left atrial myxoma resection 12 years earlier	DSA—fusiform aneurysms of the left anterior cerebral artery, the peripheral branches of the right middle cerebral artery and a giant aneurysm of the basilar artery	open heart surgery	conservative
36.	Saffie et al. [41]	37-year-old man	photopia and headache	left atrial myxoma resection 20 months earlier	DSA—left and right PCA aneurysm	open heart surgery	resection/bypass and clipping
37.	Santillan et al. [42]	68-year-old man	transient ischemic attack	left atrial myxoma resection 14 years earlier	DSA—multiple, fusiform intracranial aneurysms in the anterior and posterior circulation	open heart surgery	conservative
38.	Sato et al. [43]	64-year-old man	right arm weakness and dysarthria	left atrial myxoma recognized at the same time	DSA- multiple, intracranial aneurysms in the anterior and posterior circulation	open heart surgery	conservative
39.	Sedat et al. [44]	50-year-old woman	left hemiplegia	left atrial myxoma resection 5 years earlier	DSA—multiple fusiform aneurysms on the middle, anterior, and posterior cerebral arteries	open heart surgery	radiation
40.	Sorenson et al. [45]	53-year-old man	subacute aphasia and hemiparesis	left atrial myxoma resection 5 years earlier	DSA—multiple intracranial aneurysms, giant fusiform aneurysm of the left middle cerebral artery	open heart surgery	coil embolization
41.	Sriwastara et al. [46]	30-year-old woman	severe right sided headache, weakness of left upper and lower limbs and deviation of angle of mouth to right side with slurring of speech	left atrial myxoma recognized at the same time	CTA saccular aneurysm arising from M2 segment of right MCA	open heart surgery	cerebral aneurysm clipping
42.	Stock et al. [47]	22-year-old woman	none	left atrial myxoma resection 11 years earlier	DSA—aneurysms in both middle cerebral arteries (MCA) and right anterior cerebral artery (ACA)	open heart surgery	conservative
43.	Sveinsson et al. [48]	19-year-old woman	episodic loss of consciousness and right-sided weakness	left atrial myxoma recognized at the same time	DSA—large number of distal well-demarcated fusiform aneurysms	open heart surgery	conservative
44.	Tamuleviciute et al. [49]	29-year-old woman	TIA-like symptoms	left atrial myxoma resection 12 years earlier	DSA—multiple small and fusiform distal aneurysms	open heart surgery	conservative
45.	Vontobel et al. [50]	41-year-old woman	dizziness	left atrial myxoma recognized at the same time	MRA—multiple fusiform aneurysms	open heart surgery	chemotherapy
46.	Waliszewska-Prosół et al. [51]	62-year-old woman	vertigo, tinnitus, headache	left atrial myxoma resection 12 years earlier	DSA—multiple fusiform aneurysms located on peripheral branches of middle (MCA), anterior (ACA), and posterior (PCA) cerebral arteries	open heart surgery	conservative
47.	Waliszewska-Prosół et al. [51]	48-year-old man	a first generalized seizure due to intracranial parenchymal bleeding	left atrial myxoma resection 6 years earlier	SWI—area of intracranial bleeding in the left parietal lobe from a ruptured aneurysm; DSA—multiple fusiform aneurysms located on peripheral branches of the middle, anterior, and posterior cerebral arteries	open heart surgery	conservative
48.	Walker et al. [52]	60-year-old woman	two week history of progressive occipital headache, intermittent visual changes, right facial pain, and imbalance	left atrial myxoma resection 6 years earlier	DSA—large irregular fusiform aneurysms of the proximal SCA bilaterally and a peripheral fusiform aneurysm of a distal posterior right middle cerebral artery branch	open heart surgery	a right pterional craniotomy was undertaken but any component of the aneurysm was suitable for clipping
49.	Wan et al. [3]	39-year-old woman	headache associated with blurred vision	left atrial myxoma resection 1 years earlier	MRA—multiple aneurysms on the bilateral anterior cerebral artery, middle cerebral artery, right posterior cerebral artery and superior cerebellar artery	open heart surgery	clipping of the left ACA arterial aneurysm
50.	Xie et al. [6]	41-year-old man	abnormal behavior and logorrhea	left atrial myxoma recognized at the same time	CTA—large number of cerebral aneurysms mostly on the distal branches of both sides of middle and anterior cerebral artery	open heart surgery	conservative
51.	Xu et al. [53]	46-year-old woman	sudden anesthesia of right upper limb, paroxysmal headache for three months	left atrial myxoma resection 3 years earlier	DSA—multiple fusiform cerebral aneurysms mostly on the middle and some on anterior artery	open heart surgery	conservative
52.	Yilmaz et al. [54]	38-year-old woman	headache, episodes of right sided weakness	left atrial myxoma resection 25 years earlier	DSA—multiple fusiform aneurysms on both carotid artery territories, one of which was a giant aneurysm on the left MCA	open heart surgery	coil embolization of the giant aneurysm
53.	Yoo et al. [55]	20-year-old woman	without symptoms,4 years earlier transient left side motor weakness	left atrial myxoma recognized at the same time	DSA—multiple fusiform cerebral aneurysms, a right MCA fusiform aneurysm was the largest	open heart surgery	M2-M2 bypass surgery
54.	Zeng et al. [56]	60-year-old woman	blunt headache since 2 days	left atrial myxoma resection 2 years earlier	CTA—multiple fusiform aneurysm	open heart surgery	aneurysm was clipped after thrombus dislodgement and angioplasty
55.	Zhang et al. [57]	38-year-old woman	10 days history dizziness and headache	left atrial myxoma and aneurysm recognized at the same time	CTA—two fusiform aneurysms on the left anterior cerebral artery and left posterior cerebral artery	open heart surgery	conservative

CTA—computed tomography angiography; DSA—digital subtraction angiography; MRI—magnetic resonance imaging; CT—computed tomography; MRA—magnetic resonance angiography; SWI—susceptibility weighted imaging; MCA—middle cerebral artery; PICA—posterior inferior cerebellar artery; ICA—internal carotid artery; PCA—posterior cerebral artery; ACA—anterior cerebral artery.

**Table 2 jpm-13-00008-t002:** The most common clinical symptoms.

Clinical Presentation	*n* (%)	Women:Men	Mean Age
vascular incidents	20 (36.3)	12:8	44.5
headache	15 (27.3)	11:4	40.2
seizures	9 (16.4)	3:6	37.6
vertigo/dizziness	8 (14.5)	6:2	40.1
loss of consciousness	4 (7.3)	4:0	48
subarachnoid hemorrhage	3 (5.5)	3:0	58.3
no symptoms	2 (3.6)	-	-

**Table 3 jpm-13-00008-t003:** Location of brain aneurysm.

Location	*n*	%
multiple—the entire area of vascularization	17	31.1
MCA	16	29.1
MCA + ACA	7	12.7
MCA + PCA	5	9.1
PCA	5	9.1
ACA + PCA	3	5.5
BA	2	3.6

MCA—middle cerebral artery; PCA—posterior cerebral artery; ACA—anterior cerebral artery; BA—basilar artery.

## Data Availability

The data presented in this study are available upon request from the corresponding author. The data are not publicly available.

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
