# Peer review of "Cerebral Aneurysms Caused by Atrial Myxoma—A Systematic Review of the Literature"

_jpm, 2022, doi:10.3390/jpm13010008_

Round 1

Reviewer 1 Report

Interesting paper looking at aneurysm development in myxoma patients. 

Important topic with well done systematic review. 

The paper would further benefit from discussion regarding genetics PMID: 34863053 and role in pediatric patients PMID: 35557982.

If the above concepts addressed and references included, paper could be of interest.

Author Response

Dear Editor,

Dear Reviewers,

Thank you for your efforts in processing our paper. We greatly appreciate reviewers’ thoughtful comments, which we believe have helped us to improve the paper substantially. Below we provide a point-by-point response to these comments. All changes in the manuscript have been marked with a track-changes mode.

Marta Waliszewska-Prosół

Reviewer #1: The paper would further benefit from discussion regarding genetics PMID: 34863053 and role in pediatric patients PMID: 35557982.

Response: Thank you for this comment. We have added as suggested such a passage in the discussion based on your suggested literature (refs. 62-64)

Reviewer 2 Report

I think that the authors make a complete review about cases reported in the literature having multiple myxomas, but it is hard to find a clear conclusion about this work.  I think that authors should explain better what their article's contribution to the topic is. I have some comments:

Line 66: please rewrite the sentence

Line 67: are illustrated better than presented to avoid repeating the same word

Line 67: 'type case series' or 'a type case serie' instead of 'a type case series'

Author Response

Dear Editor,

 Dear Reviewers,

Thank you for your efforts in processing our paper. We greatly appreciate reviewers’ thoughtful comments, which we believe have helped us to improve the paper substantially. Below we provide a point-by-point response to these comments. All changes in the manuscript have been marked with a track-changes mode.

Marta Waliszewska-Prosół

Reviewer #2: I think that authors should explain better what their article's contribution to the topic is. I have some comments:

Line 66: please rewrite the sentence

Line 67: are illustrated better than presented to avoid repeating the same word

Line 67: 'type case series' or 'a type case serie' instead of 'a type case series'

Response: Thank you for this comment. We have corrected the text according to your comments and added a summary of our work. 

Reviewer 3 Report

The authors discussed  us the association between cerebral aneurysms and left atrial myxoma. Very good work

Author Response

Dear Editor,

 Dear Reviewers,

Thank you for your efforts in processing our paper. We greatly appreciate reviewers’ thoughtful comments, which we believe have helped us to improve the paper substantially. Below we provide a point-by-point response to these comments. All changes in the manuscript have been marked with a track-changes mode.

Marta Waliszewska-Prosół

Reviewer #3:

Thank you very much!

Round 2

Reviewer 1 Report

Accept

Reviewer 2 Report

The authors have responded satisfactorily to all my comments. I think that the article is publishable in his present form.